# Particles in Exhaled Air (PExA): Clinical Uses and Future Implications

**DOI:** 10.3390/diagnostics14100972

**Published:** 2024-05-07

**Authors:** Thomas Roe, Siona Silveira, Zixing Luo, Eleanor L. Osborne, Ganapathy Senthil Murugan, Michael P. W. Grocott, Anthony D. Postle, Ahilanandan Dushianthan

**Affiliations:** 1General Intensive Care Unit, University Hospital Southampton NHS Foundation Trust, Southampton SO16 6YD, UK; thomas.roe@uhs.nhs.uk (T.R.); mike.grocott@soton.ac.uk (M.P.W.G.); 2Perioperative and Critical Care Theme, NIHR Southampton Biomedical Research Centre, University Hospital Southampton NHS Foundation Trust, Southampton SO16 6YD, UK; siona.silveira@soton.ac.uk (S.S.); hings.luo@soton.ac.uk (Z.L.); a.d.postle@soton.ac.uk (A.D.P.); 3Clinical and Experimental Sciences, Faculty of Medicine, University of Southampton, Southampton SO16 6YD, UK; 4Optoelectronics Research Centre, University of Southampton, Southampton SO17 1BJ, UK; e.l.osborne@soton.ac.uk (E.L.O.); smg@orc.soton.ac.uk (G.S.M.)

**Keywords:** PExA, particles in exhaled air, asthma, acute respiratory distress syndrome

## Abstract

Access to distal airway samples to assess respiratory diseases is not straightforward and requires invasive procedures such as bronchoscopy and bronchoalveolar lavage. The particles in exhaled air (PExA) device provides a non-invasive means of assessing small airways; it captures distal airway particles (PEx) sized around 0.5–7 μm and contains particles of respiratory tract lining fluid (RTLF) that originate during airway closure and opening. The PExA device can count particles and measure particle mass according to their size. The PEx particles can be analysed for metabolites on various analytical platforms to quantitatively measure targeted and untargeted lung specific markers of inflammation. As such, the measurement of distal airway components may help to evaluate acute and chronic inflammatory conditions such as asthma, chronic obstructive pulmonary disease, acute respiratory distress syndrome, and more recently, acute viral infections such as COVID-19. PExA may provide an alternative to traditional methods of airway sampling, such as induced sputum, tracheal aspirate, or bronchoalveolar lavage. The measurement of specific biomarkers of airway inflammation obtained directly from the RTLF by PExA enables a more accurate and comprehensive understanding of pathophysiological changes at the molecular level in patients with acute and chronic lung diseases.

## 1. Introduction

Chronic respiratory diseases affect around 7.4% of the worldwide population, and acute and chronic respiratory diseases affect roughly 20% of the UK population, making this a leading cause of morbidity and mortality among adults [1,2]. Respiratory pathology is increasingly represented in acute hospital admissions, rising at three-times the rate of other medical admissions in the last decade [1]. The COVID-19 pandemic is an example of an acute respiratory tract infection that led to a significant number of hospitalisations and intensive care unit (ICU) admissions, causing a severe strain on resources [3,4]. Hospitalisations relating to acute respiratory diseases requiring oxygen therapy and advanced life support measures impose a substantial health and economic burden [5]. However, detailed assessments of acute respiratory conditions require access to distal lung samples, which is not straightforward in critically ill patients and often requires invasive procedures such as bronchoalveolar lavage (BAL) to evaluate disease mechanisms.

Small airways (<2 mm diameter) play a crucial role in the development of several respiratory diseases. For instance, increased small airway inflammation is a characteristic feature of poor disease control, leading to asthma exacerbations [6]. However, the assessment of distal small airways in clinical practice is not straightforward, and the variables used to evaluate disease progression or guide treatment decisions are usually non-specific systemic or indirect markers of inflammation. In the case of acute respiratory distress syndrome (ARDS), the pathological process of diffuse alveolar damage may occur in the days leading up to the requirement of invasive mechanical ventilation. However, there are no means of detecting patients at risk of developing ARDS in advance, and once developed, the lung damage from ARDS is often refractory to therapeutic interventions beyond lung-protective strategies [7].

Pulmonary surfactant is a complex biomolecular layer consisting a mixture of lipids and proteins that is essential for alveolar patency and adequate gas exchange. The lipid composition primarily consists of phospholipids in the form of phosphatidylcholine (PC) (80–85%), followed by phosphatidylglycerol (10%). The disaturated dipalmitoyl-PC (DPPC) is the principal PC involved in alveolar surface reduction accounting for most of the PC composition (40–60%) [8]. Surfactant proteins (SPs) account for around 10% of the composition, the most abundant of which is SP-A [9]. SP-A and SP-D are essential for innate immune system, and SP-B and SP-C are involved in surfactant adsorption [9,10]. Lung inflammation can lead to increased alveolar permeability due to breach in the epithelial–endothelial barrier and subsequent invasion of plasma constituents such as albumin and haemoglobin. This can lead to a compromise in the surface tension-reducing ability of the alveolar surfactant. Surfactant deficiency from insufficient surfactant synthesis/secretion, increased breakdown (either by hydrolysis or oxidation), or surfactant inactivation/inhibition by biophysical inhibitors can lead to increased alveolar surface tension and poor lung compliance [11]. The pulmonary surfactant is synthesised, secreted, and recycled by alveolar type II (AT-II) cells, and during acute lung injury, there are significant compositional changes in both phospholipid and protein fractions of the respiratory tract lining fluid (RTLF). Since the composition of RTLF reflects the composition of the surfactant, monitoring its composition can be an early indication of respiratory deterioration before physical symptoms appear. Dysfunctional surfactant (in composition, quantity, or function) can be used as markers of small airway pathology including asthma and chronic obstructive pulmonary disease (COPD) [12,13,14].

## 2. Sampling the Respiratory Tract Fluid Lining

The RTLF can be accessed by different methods. Induced sputum is a non-invasive sampling method where airway sample is induced by coughing after inhalation of isotonic or hypertonic saline [15]. Induced sputum is easily reproducible in the clinical context and allows microbiological and cellular compositional analysis through various analytical methods. However, there are limitations of this sampling method. For example, the induced sputum may be contaminated with secretions from the oral cavity and upper respiratory tract, leading to inaccurate results. Additionally, there may be variations in the quantity of sputum production and cellularity of samples depending on the origin of the material. Furthermore, induced sputum composition may not be reflective of deep RTLF, as the material is primarily from the central airways and consequently has limitations in accessing alveolar and distal small airway diseases [16,17].

Sampling RTLF from the distal airways is also possible through bronchoalveolar lavage (BAL). A bronchoscope, a thin, flexible tube, is passed through the nose or mouth (in the ICU, usually via an endotracheal tube) and into the airways. The bronchoscope is guided through the trachea and into the segmental and subsegmental bronchi. Sterile saline solution is injected through the bronchoscope and into a specific part of the lung. The saline is then gently suctioned back, washing the alveolar tissue, and collecting a sample of the BAL fluid. BAL samples can be analysed for components similar to that of induced sputum, however given that the samples are obtained from more distal (smaller) airways, additional assays can be performed including cytokine differentiation, cellular infiltration, surfactant products, microbial material, and endothelial surface markers [18]. However, this is an invasive procedure and requires sedation, which may impose additional complications with resource utilisation. Moreover, it suffers with problems relating to the unknown dilution factor of the sample by the lavage fluid itself and potential contamination of the sample due to airway contact bleeding [19]. The samples are often taken from dedicated areas and may not be reflective of the global lung picture. Whilst BAL fluid offers samples closest to the site of insult, non-invasive sampling would be better suited to minimise procedural complications in the clinical environment. Plasma or serum are alternatives and may provide easy access to the systemic vascular circulation and its components. However, although these may help to measure the leakage material from the lungs, they are not reflective of the pathology at the alveolar level.

A non-invasive alternative method to extract deep lung samples is exhaled breath condensate (EBC). The European Respiratory Society (ERS) and American Thoracic Society (ATS) Task Force (TF) defines EBC as exhaled breath cooled to a fluid or frozen using a condenser [20,21]. In general, sample collection involves a patient inhaling and exhaling into a mouthpiece connected to a tube made from a chemically inert material, which feeds the air to the condenser, where particles grow by the condensation of water vapor and deposit [22]. Exhaled breath is an aerosol of gases and particles with diluted non-volatiles, such as surfactant lipids and proteins including cytokines, and volatile water-soluble compounds including hydrogen peroxide and carbon dioxide [22,23]. Small droplets mainly originate from the small airways during airway closure and opening [21]. When the temperature of the water vapor falls below the dew point, the aerosol condenses into large droplets on the condenser surface [23]. This method can be used to collect samples from ventilated or non-ventilated subjects repeatedly, with studies on diseases including asthma, ARDS, pneumonia, and COPD performed previously [22,24]. However, the composition of EBC is very sensitive to collection factors including ambient temperature and humidity, which can result in altered content and concentration of EBC components. Furthermore, the concentration of volatile dissolved compounds such as hydrogen peroxide and carbon dioxide has been found to decrease following the collection, so measurement of such compounds must be performed as soon as possible. EBC is also subject to contamination, in particular due to the potential inclusion of samples from the upper airways or oral cavity. Salivary traps are a solution for oral contamination, however ensuring specificity of the sample to particular regions of the airway is difficult [22]. Finally, there is a lack of standardisation, with intraindividual variability and differing collection equipment posing challenges [22,25]. The EBC method is not designed for the collection of particles, the fraction that contains non-volatiles such as lipids and proteins. Collection efficiency for small particles by condensation is often inefficient with a large and variable dilution with water [26]. In comparison, the PExA device counts the number and size of particles collected, providing a method to normalise components and thus reducing issues of intraindividual variation [25].

Additionally, exhaled breath contains a wide variety of volatile organic compounds (VOCs) produced during biological processes [27]. Changes in VOCs detected may indicate abnormal biological processes; for example, they have shown promise as biomarkers for a wide variety of diseases including Alzheimer’s, ventilator-associated pneumonia, and lung cancer [28,29,30]. Like EBC, such a method is non-invasive, however VOCs in exhaled breath tend to be trace quantities (usually concentrations of parts per billion (ppb)) [27]. Therefore, collection is sensitive to environmental contamination or variation in temperature causing degradation of molecules. Being volatile, sample preparation and storage is challenging due to decomposition, so analysis techniques must be carefully selected; e-noses, such as gas sensor arrays, are one potential method, offering portability [30]. Mass spectrometry is often used, as it can provide a low limit of detection down to ppb; however, there is a potential for sample degradation during the process due to increased temperature [27]. Thus, whilst VOCs offer a disease-specific and sensitive detection method, care must be taken during sample collection, preparation, and analysis, requiring expert users.

Other methodologies to assess small airways non-invasively, including inert-gas washout techniques and impulse oscillometry, merely reflect small airway structure, do not provide a biological sample, and have limitations (including intraindividual variability, contamination issues, and detection limitations of assays) that prevent their widespread use in standard clinical practice [21,31]. Clearly, there is a need for a reliable, non-invasive, reproducible, and easily quantifiable technique for monitoring small airway pathology to facilitate both the diagnosis and prognostication of acute and chronic respiratory diseases in the clinical setting.

## 3. PExA Basics

The analysis of droplet particles in exhaled air (PExA) meets the demand outlined above by offering a viable means of RTLF biochemical analysis that is non-invasive and reproducible [32]. Aerosolised particles produced by breathing are collected in the instrument via impaction on a membrane, enabling sampling of the distal lung environment non-invasively [33]. The breath aerosol is drawn through nozzles in the impactor that direct the airflow towards an impactor plate that forces the air to deflect into streamlines around it (Figure 1B). Large particles exceeding the impactor cut-off size are unable to follow the air streamlines and will impact onto a thin membrane, whereas small particles are diverted by the air streamlines around the impactor plate. The impactor is designed with several stages where the nozzle dimensions are reduced for each consecutive stage to increase the velocity of the air and to collect a smaller particle fraction. PExA samples (often abbreviated to PEx) can be subsequently analysed for various biomolecules including cytokines, proteins, miRNA, and phospholipids. An optical particle counter (OPC), connected just before the impactor, draws a small fraction of the aerosol to measure particle size and number concentrations. The OPC data are used to calculate the collected particle mass online to enable collection of a predetermined particle mass. Since the number of exhaled particles show a large variation per litre of exhaled breath, it is highly beneficial for the downstream chemical analysis to standardise the collection to a selected particle mass to have all samples in the optimal concentration range for the analysis. When the sampled mass of particles is known, the analyte concentration in the undiluted RTLF can also be determined.

The selection for particles produced in the small airways has been established by using particle size limitation and using a standardised breathing manoeuvre [12,34]. The particle flow rate produced by the patient is increased by utilising breathing manoeuvres that promote airway closure and reopening (breathing to residual volume, breath hold, inhalation to total lung capacity, and further exhalation to residual volume). This indicates that the smaller collapsible airways are the main origin sites for particle production [35]. Furthermore, Behndig et al. analysed PExA, BAL, and bronchial wash samples of 15 healthy volunteers to determine the relationships between PExA sampling and more traditional sampling techniques [36]. They noted a significant association of exhaled albumin between PExA and BAL samples (r: 0.65, *p* = 0.01), but not between bronchial wash and PExA samples. After correction for BAL dilution, SP-A was also strongly correlated between PExA and BAL samples (r: 0.61, *p* = 0.015) [36]. Given BAL and bronchial wash specimens provide samples of the small and large airways, respectively, this study highlights the capability of PEN to reliably and non-invasively access the small airways in humans.

In addition to particle flow rate, samples can be analysed through mass spectrometry for lipidomics or enzyme-linked immunosorbent assay (ELISA) for proteomics to establish the chemical composition of the small airways, allowing differences to be established between physiology and pathology [19]. Particles sampled and assessed through PExA have considerable potential; Bredberg et al. (2012) identified 120 different proteins that can be detected, with up to 80% being concurrently detected via invasive bronchoscopy, and Östling et al. identified 207 proteins from PExA samples, with phenotypic distributions depending on the smoking status [37]. An additional advantage of using PExA (in comparison to both BAL and tracheal aspirate) is that it does not involve administering foreign substances into the respiratory system that can alter or dilute the respiratory fluid composition [13]. PExA can be modified (PExA 2.0, see Figure 2) to connect directly to the expiratory circuit to collect samples from mechanically ventilated patients [38].

Exhaled particles are predominantly surfactant material, made up of phospholipids and proteins. Di-palmitoyl-phosphatidylcholine (DPPC) is the most frequently observed lipid, which functions predominantly to maintain airway patency. SP-A has a pivotal role in immunological defence to pathogens, binding directly to foreign material that is too small for alveolar antigen-presenting cells. The interplay between DPPC and surfactant proteins, including SP-A, maintains the physiological alveolar lining. Other proteins that are detected include immunoglobulins and albumin. The disturbance of the biochemical changes during disease processes has been explored and is discussed later in this review [19,32]. Koca et al. (2022) analysed PExA and blood SP-A samples in healthy people (non-smoker, non-asthmatic, FEV1 > 80% predicted) and smokers and identified that PExA SP-A was significantly lower in smokers compared to that in healthy participants, and the ratio of plasma-PExA SP-A was significantly higher in smokers compared to that in healthy participants [39]. Given that there was no correlation between PExA and plasma SP-A in healthy participants, this study implicates the possibility of SP-A being used as a marker of alveolar permeability.

The physiology of the surfactant must be considered and appreciated before analysis of PExA can be performed. The concentration of exhaled particles increases with age but there is a negative correlation between increasing age and DPPC concentration [19]. As mentioned, DPPC acts to prevent alveolar closure, meaning the reduction induces more collapse and subsequent reopening, leading to more particle exhalation. The study also demonstrated a link between increases in DPPC concentration and FEV1/FVC ratio, highlighting the role of these phospholipids in healthy lung function. Interestingly, increased concentration of DPPC was also associated with smoking, which suggests that it is not just low levels that are pathological [19]. Fessler and Summer (2016) concluded that increased DPPC may in fact induce phospholipid dysfunction, leading to increased alveolar collapse and subsequently elevated uptake on the PExA device [40].

Interindividual variability can be high for PExA, with Bake et al. (2017) determining that such a variation is explained only in part (28–29%) by age, anthropometric data, and spirometry variables; the rest remains poorly understood [12]. Intraindividual variation appears to be dependent on the time of day the samples are taken [41]. In 16 healthy volunteers, SP-A and albumin were measured using a PExA device in the morning, noon, and in the afternoon. From morning to noon, there were significant increases in both SP-A and albumin levels, with non-significant difference between noon and afternoon samples. Intraindividual variability was not shown to be dependent on the days between PExA use, indicating that the variability was not due to user familiarity [41]. Furthermore, the diurnal variability of these exhaled biomarkers appears to coincide with the serum markers of lung epithelial injury (SP-D and Club-Cell protein 16 (CC-16)), highlighting a time-dependent production and secretion of surfactant proteins or differences in epithelial tight-junctions resulting in transepithelial leakage [42,43].

## 4. PExA Techniques

### 4.1. Breathing Manoeuvre

The standardised breathing manoeuvre, as described by Bake et al. and Östling et al., requires a full exhalation to residual volume followed by breath holding for five seconds followed by maximum inhalation to total lung capacity and normal exhalation to functional residual capacity [12,37]. This gives rise to a release of high numbers of tiny droplets/particles formed from the respiratory tract lining fluid covering the small airways in lungs [12,37]. In mechanically ventilated patients, continued measurements can be made via the PExA 2.0 device, where the expiratory limb of the circuit from the patient is attached to the PExA device prior to connecting to the ventilator as protocolised and demonstrated by Broberg et al. and graphically depicted in Figure 2 [38].

### 4.2. Analytical Techniques

As a promising tool for quantifying specific proteins in biological samples, ELISA has been commonly used. SP-A and albumin in PExA samples were analysed using ELISA kits [41,44], in comparison to that in BAL and bronchial wash samples [36], as well as to other diagnostic indicators of respiratory diseases including fractional exhaled nitric oxide (FeNO), exhaled breath temperature (EBT), and exhaled VOCs [45].

Following extraction with the PExA machine, PEx particles can provide a comprehensive proteomic profile. Östling et al. evaluated the performance of SomaScan (SomaLogic, Boulder, CO, USA), a proteomics platform based on DNA aptamers, for PExA analysis in asthma patients measuring the expression of more than 1100 proteins [37]. A total of 32 proteins were identified with differential expression between asthma and control groups. Of interest was also that proteins belonging to the complement system seem to be associated with asthma control [46]. SomaScan was also used to explore the effect of smoking by Kokelj et al., and 203 different proteins were detected, and 81 proteins differed between the groups, where a difference depending on sex was apparent [47]. The results suggested the potential of PExA proteomics in revealing novel biomarkers and pathways involved in airway inflammation pathogenesis.

Several studies have used liquid chromatography–mass spectrometry (LC-MS) to characterise the protein and phospholipid composition of PExA, suggesting similar composition to BAL samples and to a lesser extent to induced sputum (ISP) [48]. Surfactant from small airways will be transported to glottis, therefore all three sampling methods (BAL, PExA, and ISP) have similar surfactant but is collected at different places in the airway. There are minor differences that suggest that PEX could be a sample that is more selective to collecting the lung surfactant in small airways, for example, the very low content of SM lipid [48].

Holz et al. (2022) demonstrated the feasibility of using Meso Scale Discovery (MSD, Svar Life Science, Malmö, Sweden) platform, a multiplex immunoassay system to detect the levels of inflammatory cytokines including interleukin (IL)-6, IL-8, and myeloperoxidase (MPO) and proteins (SP-D, albumin) in PExA samples following segmental and inhalation of endotoxin (LPS) challenge in healthy volunteers [25]. Following segmental challenge, there was an increase in the particle emission and PEx concentration, but the particle size distribution was not changed. Despite being a segmental challenge rather than whole lung inflammation, there were significant increases in IL-6 and IL-8 detected in PEx. Although there was no significant difference in SP-D after both (segmental and inhalational) challenges, it was detectable at PEx mass concentrations of 120 ng [25]. All analytical techniques described follow protocolised methodologies to ensure standardisation of results.

### 4.3. Maintenance Requirements of PExA Machine

The minor maintenance aspect that should be considered with the PExA instrument is the potential for sample contamination by the exhaled droplets with previous samples. This is minor given the extremely low sample material that is exhaled. An ordinary sample is 100 ng of material. This means that it takes 10 million sample collections of PEx before one gram has been exhaled into the instrument. The droplets dry down and adhere to the walls where they remain indefinitely. This is confirmed by the particle data that show that particles in the instrument are not collected when breath sampling has ceased.

One must also ensure that the sampling membrane never touches any surface in the impactor or the instrument (Figure 1B). After collection, the sample is stamped or cut out from the centre of the sampling plate. The impactor nozzles (shown in Figure 1C) experience high air velocities, and as such, intermittent cleaning by flushing with methanol is required. If there is an excessive build-up of particles around the nozzles, there is potential risk of cross contamination.

## 5. PExA in Pathology

PExA usage in various pathological settings has been investigated. These are summarised in Table 1 and are discussed in more detail here.

### 5.1. Smoking

Smoking influences the concentration of surfactant phospholipids detected in PEx, where both increases and decreases in surfactant concentrations were noted [19,37]. This result is supported by Viklund et al., wherein increased DPPC (and palmitoyloleoyl PC (POPC)) was detected in PExA of current smokers compared to that in never-smokers, even when lung function was within the normal range in both groups [65]. Interestingly, SP-A was increased in current smokers compared to never-smokers, but in those with normal lung function, SP-A levels were non-distinguishable between smokers and non-smokers. Possibly, this is a reflection of the role of SP-A in the immunological response to foreign material in the lungs, therefore raised SP-A in this scenario is a marker of ongoing toxin exposure rather than decreased lung function. Smoking induces lung inflammation and recruitment of macrophages, which in turn can lead to increased surfactant synthesis and metabolism by type II pneumocytes and macrophages as a compensatory process. As the damage exceeds this compensation, the deterioration in lung function via spirometry becomes apparent [67]. This, in turn, leads to increased alveolar collapse and therefore increased PExA particle mass. This was demonstrated in the results of the study by Viklund et al., wherein lung function deterioration in smokers was associated with significantly increased PExA sample mass [65]. Given that lung function deterioration in smoking often occurs later then the development of structural damage (including airway thickening and emphysema), PExA offers a potential means of detecting early lung damage [68].

There are 81 proteins in PEx shown to differ significantly between smokers and never-smokers [47]. The largest differences were observed for e sRAGE, FSTL3, SPOCK2, and protein S, all of them being less abundant in current smokers. Circulating sRAGE has been shown to be decreased in COPD and is suggested to reflect a marker of deficient inflammatory control [69,70]. Other findings of note were the differences in proteins belonging to the complement system in smokers, where a marked difference in response to smoking was observed between men and women. Moreover, 62 proteins were less abundant in current smokers (*n* = 38) compared to ex-smokers (*n* = 47), and four of those remained less abundant in ex-smokers than in never-smokers (*n* = 22) (MRC1, CD55, ST2, AK1) [47].

### 5.2. Asthma

Asthma is a chronic respiratory condition characterised by airway inflammation resulting in airway narrowing and persistent symptoms. Clinical phenotypes vary, and traditional investigative methods involve measuring peak expiratory flow, blood eosinophil count, spirometry, bronchial provocation tests, and exhaled nitric oxide (FeNO). While these tests help assess the degree of airflow limitation and airway inflammation, they do not provide molecular-level information concerning the changes at the small airway level [71]. However, PExA has the potential to comprehensively evaluate specific markers, such as proteins, lipids, metabolites, cytokines, and other markers that are associated with airway inflammation from the RTFL. The non-invasive nature of the particle capture may enable diagnosis, disease monitoring, or treatment response [45].

Reduced alveolar reopening is a sign of small airway dysfunction, and reduced exhalation of PExA particles and PEx mass is a typical feature of patients with asthma. Moreover, persistently decreased PEx mass was noted in asthmatic patients (*n* = 18) when compared with those with complete remission (*n* = 12) and healthy controls (*n* = 18) and associated with small airway hyper-responsiveness [56]. Furthermore, increased PEx mass was associated with reduced broncho-alveolar hyper-responsiveness and more favourable FEV1, FVC, and PEFR and reduced RV. This suggests that the quantification of PEx mass may differentiate between healthy people and patients with asthma, even those in clinical remission. These findings were further confirmed by additional exposure tests, whereby reduced particle mass was noted after birch pollen exposure during pollen season in subjects with birch pollen allergy and mild asthma [13].

Although this birch exposure study (during the pollen season) did not show any significant changes in the concentrations of SP-A and albumin within the PEx samples, Soares et al. demonstrated that there appears to be specific SP-A profile within 83 asthmatic patients of varying severity [44], meaning PExA has the potential to sub-phenotype asthma patients into those with and without small airway hyper-responsiveness. The study highlighted an association between lower percentage SP-A levels in patients with small airway inflammation and with reduced FVC compared to those in the healthy volunteers, suggesting that reduced SP-A levels have a causal relationship with small airway dysfunction and subsequent airway closure [44].

Alahmadi et al. assessed the short- and long-term effects of inhaled corticosteroids (ICS) in patients with asthma with an FeNO of ≥45 ppb and demonstrated that increased percentage predicted FEV1 was strongly correlated with increased PEx mass and particles exhaled per breath, two hours after the administration of ICS [45]. However, this association was not demonstrated after seven days of regular use of ICS. Although this was a small study and only consisted of 10 participants who were able to perform PExA, there was a trend towards negative association between FeNO and PEx (ng) at the baseline. Moreover, the PEx mass correlated significantly with the levels of SPA on day 1. This was thought to be a likely consequence of both increased air trapping preventing small airway closure in more severe disease and more proximal (larger airway) involvement in the inflammatory response seen in asthma [45,72]. However, improvements after ICS were only seen in FeNO, exhaled breath temperature (EBT), and six selected VOCs, and there were no dynamic changes in PEx measurements [45]. More recently, an exploratory study showed that complement and coagulation markers in RTLF are associated with asthma control and small airway dysfunction, highlighting the potential contribution of the complement cascade in the pathogenesis of asthma [46].

Surfactant phospholipid and protein measurements in asthmatics extracted from PExA showed varying results. In a study of 15 adult asthmatics when compared with controls, the ratio of unsaturated to saturated phospholipids was higher in patients with asthma [49]. The disaturated PC such as PC16:0/16:0 is thought to be more surfactant specific, while the unsaturated PC composition may reflect inflammatory cell membrane composition during airway inflammation, which may explain these findings [49,73]. In a large population-based study (*n* = 200) including healthy volunteers and an enriched sample asthmatics from Sweden, certain phosphatidylcholine composition [PC 14:0/16:0 and PC16:0/18:2] and SP-A was found to be increased in patients with asthma. Moreover, smokers had a higher fractional concentration of PC16:0/16:0 than non-smokers [19]. The observed differences are likely due to variations in the populations studied. During the early stages of lung damage, there may be an increased production of surfactant. However, as the lung function deteriorates with a proportional decline in AT-II cells, the synthesis and turnover of surfactant may be affected, leading to a deficiency of surfactant components. SP-A dysfunction has been linked to asthma in animal models, whereby SP-A deficiency following allergen exposure results in disinhibition of Th2 cellular responses, with SP-A administration acting to regulate this effect [74]. Further, this hypothesis is supported in a systematic review, where SP-A/D function is more frequently impaired in eosinophil-driven lung inflammatory conditions [75]. The alterations in surfactant composition may suggest that exogenous surfactant potentially has a role in moderating airway inflammation, and further studies are needed to evaluate the effect of exogenous surfactant in asthmatics.

### 5.3. Chronic Obstructive Pulmonary Disease (COPD)

COPD is a chronic progressive lung condition characterised by airflow obstruction secondary to partly reversible inflammation and degenerative lung damage predominantly in the small airways and lung parenchyma [76]. Assessment of RTLF may offer a non-invasive evaluation of underlying pathological processes in the small airways associated with COPD. Lärstad et al. in a small observational study noted a decrease in SP-A in PExA samples as COPD disease severity worsens (as per the GOLD severity criteria), and the difference between COPD patients (*n* = 13) and healthy subjects (*n* = 12) was stark and statistically significant [14]. However, PEx albumin levels were not significantly different between COPD and healthy subjects. The lower SP-A levels may be due to the mechanisms relating to COPD pathology, whereby chronic airway inflammation and oxidative stress lead to increased surfactant degradation with or without leakage of products into the pulmonary and systemic circulation from the alveoli. This conclusion correlates with other studies, where BAL surfactant levels in COPD are reduced, but significantly increased in induced sputum, highlighting that airway closure location is likely to be more proximal [77,78]. Alternatively, given the alteration in lung mechanics in COPD, the decreased SP-A and total PFR due to more proximal airway closure, lead to lower availability for small airway re-opening and subsequent reduced PExA sample aerosolisation. This, coupled with the reduced elastic recoil of the alveoli in COPD, leads to fewer airways being available for inclusion in PExA measurement [79].

### 5.4. Acute Respiratory Distress Syndrome

ARDS is an acute inflammatory lung disease with pathological changes of diffuse alveolar damage leading to alveolar epithelial and endothelial injury. Patients present with acute hypoxemic respiratory failure secondary to poor lung compliance and significant non-hydrostatic pulmonary oedema often needing initiation of mechanical ventilation [80,81]. ARDS is diagnosed when the patient fulfils the selective criteria called the Berlin Definition of ARDS and more recently the new Global Definition of ARDS [82,83]. The Berlin definition categorises the disease severity according to the degree of hypoxemia as defined by the ratio of partial pressure of arterial oxygen to the fractional inspired oxygen (PaO_2_/FiO_2_, mild: 200–300 mmHg; moderate: 100–200 mmHg); severe: ≤100 mmHg) [82]. There are alterations in surfactant phospholipid and protein composition, metabolism, and function, and this dysregulated surfactant metabolism is likely due to a combination of reduced surfactant synthesis, secretion, recycling, and increased breakdown and inhibition [11,84,85,86,87]. Clinical trials of surfactant replacement in ARDS demonstrated no clear survival advantage, and this is likely due to a lack of understanding of mechanisms relating to surfactant metabolism in humans with ARDS [11]. With a mortality rate around 30–50%, there is clearly a pressing requirement for detection of specific diagnostic and prognostic markers to identify respiratory deterioration prior to the onset of physical symptoms, allowing earlier treatment and potentially improved outcomes for the critically ill patient [64].

When PEx is sampled in porcine models of ARDS induced by endotracheal administration of lipopolysaccharide (LPS) directly into pulmonary arteries, compared to controls (saline), there was a significant increase in particle flow rate (PFR), peaking at 60 min post administration. This was around 30 min prior to any clinical or biochemical parameters becoming deranged in the ARDS models [64]. Segmental LPS administration in healthy adults and porcine models leads to a significant rise in particle emission and PEx mass concentration with increased inflammation [25,64]. This phenomenon in increased PFR was demonstrated during primary graft dysfunction in transplant patients [52]. These studies suggest that increased PEx PFR may potentially help to detect acute lung injury in patients developing ARDS prior to clinical deterioration.

Given the mainstay of ARDS management that has the most conclusive evidence basis is lung protective ventilation strategy, it is prudent to evaluate how invasive mechanical ventilation settings influence PExA parameters. In theory, reduced PFR indicates a “gentler” form of ventilation due to decreased re-opening and collapsing of small airways. Supporting this, Hallgren et al. suggested that PFR increases significantly when pressure support ventilation (PSV) is instigated in comparison to pressure-controlled ventilation (PCV) or volume-controlled ventilation (VCV), by analysing the PEx from 30 elective open-heart surgery patients [60]. This finding is by virtue of the increased ventilator dysynchrony, which causes more chaotic and inefficient ventilation. Studies have identified little difference in PFR between PCV and VCV [52,60]. In comparison, pressure-regulated volume control (PRVC) is associated with significantly reduced PFR compared with PCV and VCV [60]. However, in vivo porcine models suggest variations in particle flow between PCV and VCV ventilation strategies with increased particle mass and larger particle flow during high tidal volume ventilation (TV 10–12 mL/kg) [52,53].

### 5.5. COVID-19

The COVID-19 pandemic unearthed the diagnostic and therapeutic challenges facing modern medicine on a global scale, particularly in acute respiratory infections, where the limitations of airway and lung sampling meant very little was understood about the pathophysiological mechanisms when the disease first emerged.

Viklund et al. demonstrated that viral RNA of COVID-19 can be detected within five minutes of breathing into the PExA device and that the number of particles exhaled was significantly lower in COVID-19 patients versus those in controls during both normal breathing and the small airway selecting breathing manoeuvres (albeit not as sensitive as standard PCR methods of testing) [33]. Conversely, Hirdman et al. performed proteomic analysis of PExA samples of COVID-19 positive patients (*n* = 20), symptomatic patients with a negative COVID-19 test (*n* = 16), and healthy control groups (*n* = 12). They demonstrated significantly higher particles per exhalation in COVID-19 confirmed cases compared to those in healthy controls, and a slightly higher, albeit not statistically significant, particles per breath compared with symptomatic non-COVID-19 patients indicating a disease-specific trend in exhaled particles [61]. This result is in accordance with other studies investigating respiratory disease and COVID-19 respiratory disease [64,88]. Furthermore, proteomic analysis enabled profiling of PEx amongst the three groups, identifying proteins implicated in immune activation, coagulation cascade, and RTFL components [61]. COVID-19 patients produced greater ORM-1, alpha-1-antitrypsin, and haptoglobin acute phase proteins compared to both COVID-19 negative and healthy control patients [61]. ORM-1 has prognostic implications in pneumonia, providing the possibility of clinical significance of this testing methodology [89].

COVID-19, rather characteristically, appears to influence both local and systemic coagulation, alongside hyper-inflammatory responses with a dysregulated immunomodulatory effect [90]. This again was demonstrated in the proteomics of the RTLF in COVID-19 patients, where there was increased APOA1 and transferrin (which is known to influence the coagulation cascade through interference with antithrombin function and factor XIIa). Overall, this further suggests that there are disease specific pathophysiological mechanisms reflected in the RTLF, which can be successfully sampled using the PExA machine.

## 6. Future Directions

Acute and chronic lung diseases require detailed analysis of distal small airways to mechanistically characterise distinct clinical phenotypes that may respond to individualised therapies. However, the assessment of small airways requires invasive procedures; measurement of RTLF biomarkers from PEx particles provides an alternative and complementary approach to the existing investigative methods. Currently, PExA is mostly used as a research tool to evaluate mechanistic processes that underpin disease pathology in respiratory conditions, but the clinical implications can be manyfold (Figure 3). Our current understanding of the disease–host response is beginning to widen as the incorporation of multi-omics to standard clinical practice becomes more readily available. Already, in asthma and COPD, multi-omics data are facilitating phenotyping, biomarker identification, and the introduction of personalized medicine [91,92,93].

While PExA has been used as a tool mostly to characterise chronic airway diseases such as asthma, further studies are needed to assess the utility of PExA in acute respiratory conditions such as pneumonia (both viral and bacterial) and ARDS. Moreover, PExA may be useful in evaluating suitable ventilation strategies to minimise ventilator-induced lung injury in mechanically ventilated patients. In ARDS, although surfactant compositional and functional changes are characteristic, the replacement of exogenous surfactant proves no value in improving clinical outcomes. This is primarily due to the lack of understanding of mechanisms relating to surfactant metabolism and turnover following exogenous surfactant replacement. PExA may provide a rapid, continuous, and non-invasive method of surfactant characterisation, which may help refine therapeutic strategies according to surfactant metabolism in vivo.

## 7. Conclusions

PExA is a particle counter device that captures small particles from distal airways. Unlike BAL, PExA is a non-invasive tool for lung sample extraction without the need for additional sedation or saline installation. Although significant alterations in RTLF have been demonstrated in PEx particles from several lung diseases, it still remains a complementary research tool. The common molecular compositional changes are seen in phospholipids, proteins, cytokines, and miRNA levels. Exploration of molecular phenotypes by PExA may help to define specific patient populations prior to individualised treatment interventions in the future. Moreover, while PExA is currently mainly used in spontaneously breathing patients, further clinical studies are needed to evaluate the use of PExA in non-invasively ventilated and mechanically ventilated patients.

## Figures and Tables

**Figure 1 diagnostics-14-00972-f001:**
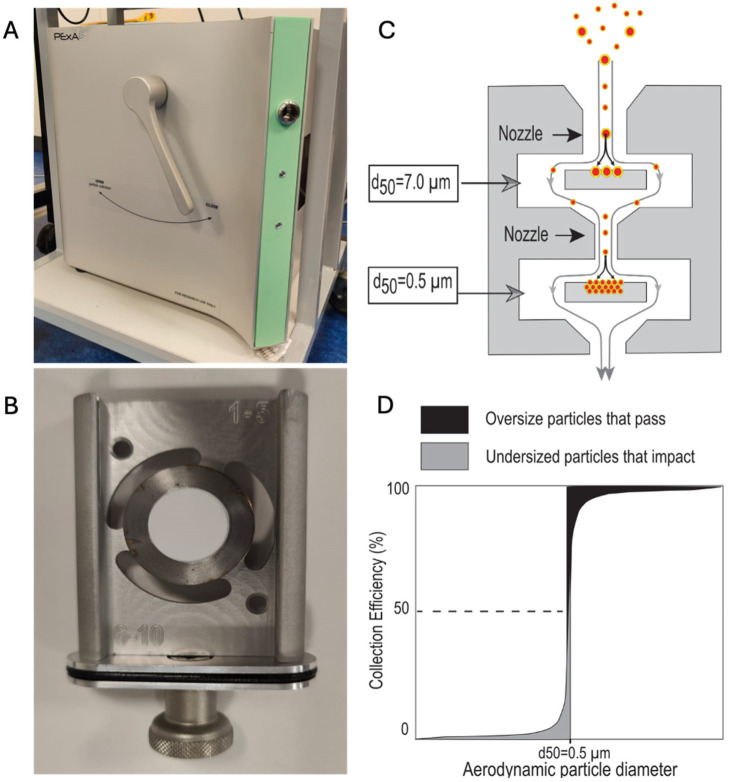
PExA device and principles. (**A**) Photograph of the external surface of PExA including the flow director level and opening for mouthpiece. (**B**) PExA impaction tray. (**C**) Schematic of PExA air flow. Note the smaller (0.5–7 μm) particles are selected by bypassing the initial tray and are impacted on the second. (**D**) Principle of a 50% particle size cut-off (PExA impactor cut-off curve is not determined).

**Figure 2 diagnostics-14-00972-f002:**
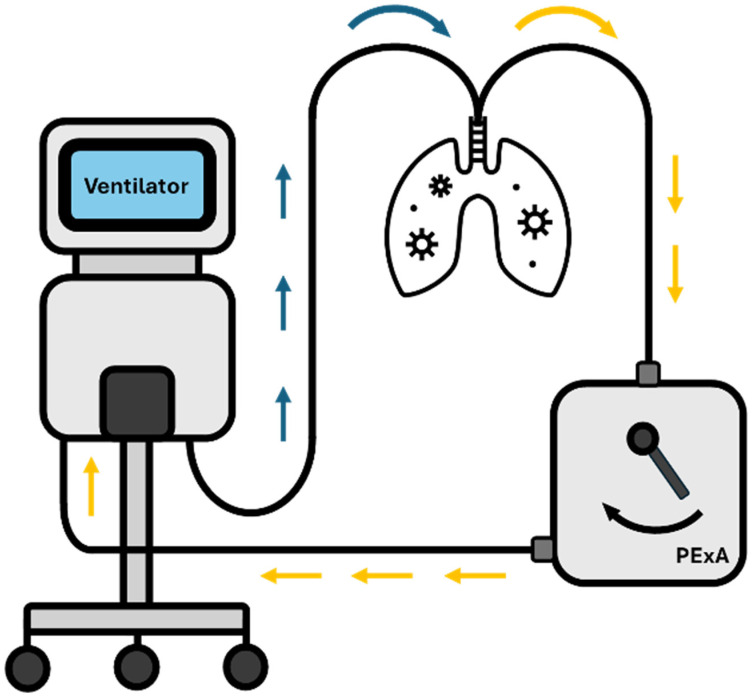
Representation of PExA in 2.0 configuration, connected to a mechanical ventilator. Blue arrows indicate inhaled air. Orange arrows are for exhaled air.

**Figure 3 diagnostics-14-00972-f003:**
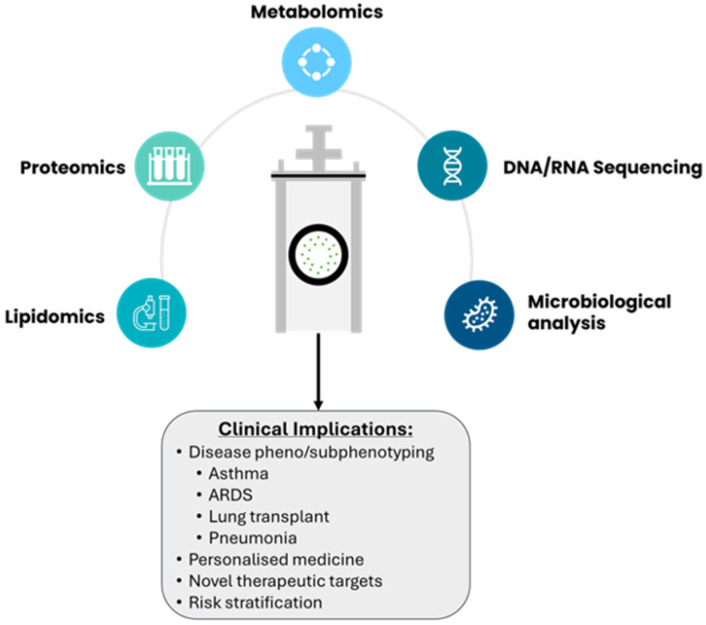
Analytical techniques utilized for PExA samples are highlighted above, with some potential clinical applications.

**Table 1 diagnostics-14-00972-t001:** Studies using PExA in their methodology to assess pathophysiological processes in disease states.

Authors	Clinical Condition	Biomarker	Outcome/Key Results
Alahmadi et al. [45]	Asthma (*n* = 17)	Exhaled breath tests, FeNO, EBT, PExA, and VOCs	(1) After a week of using high-dose inhaled corticosteroids, there were falls in FeNO, EBT, and two VOCs (*p* < 0.05), but no changes in PExA. (2) There were no significant differences in the calculated weight percentage of SP-A (*p* = 0.989) or albumin (*p* = 0.674) between day 1 and day 7 in PExA samples.
Almstrand et al. [49]	Asthma (*n* = 15)	Total particle count, phospholipid composition	(1) Subjects with asthma exhaled significantly lower numbers of particles than controls (23,000 vs. 44,000, *p* = 0.03). (2) The ratio of unsaturated to saturated phospholipids was significantly higher in samples from subjects with asthma (0.25 vs. 0.35; *p* = 0.036).
Andreasson et al. [50]	Lung adenocarcinoma (LUAD) (*n* = 17), non-cancer surgical controls (*n* = 18)	Particle flow rate, hepatocyte growth factor receptor (MET)	(1) A significantly higher particle flow rate was seen among LUAD patients before surgery compared to that in the control patients (*p* < 0.0001). (2) A significantly higher MET concentration was found before surgery in the LUAD group compared to that in the control group (*p* < 0.0001).
Bredberg et al. [51]	Smoking (*n* = 12)	Phospholipid composition	(1) Clear discrimination between smokers and non-smokers, where phospholipids from smokers were protonated and sedated to a larger extent. (2) Poor lung function showed a strong association with higher response from all molecular PC species.
Broberg et al. [38]	Non-small cell lung cancer with mechanical ventilation (*n* = 17) versus controls	Albumin and SP-A, particle flow rate	(1) Mechanically ventilated patients with non-small cell lung cancer showed significantly lower levels of DPPC in PEx samples compared to non-intubated patients (*p* = 0.001). (2) Established the feasibility of PExA device to collect and analyse exhaled particles from lung airways.
Broberg et al. [52]	Lung transplant—primary graft dysfunction (*n* = 6) and no primary graft dysfunction (*n* = 6)	C-reactive protein (CRP), particle flow profile	(1) Patients with PGD had significantly higher CRP levels after transplant on day 0 compared with patients with no PGD (*p* = 0.0420). (2) Lung transplant patients with PGD show a significant difference in total particle count between day 0 and day 1 compared with day 3 (*p* = 0.0065 and *p* = 0.0082, respectively).
Broberg et al. [53]	Porcine model—pigs (*n* = 6)	Particle flow	(1) Particle mass was significantly higher in pressure-controlled ventilation (PCV) than in volume-controlled ventilation (VCV) (*p* = 0.0322).
Broberg et al. [54]	Porcine models of mechanical ventilation (*n* = 6)	Total particle count	(1) Comparing VCV to PCV from day 1 to day 3, a significant increase in total particle count was observed on day 2 (40,260 ± 10,097 vs. 21,238 ± 5625, *p* = 0.0184), with the highest particle count occurring during VCV.
Broberg et al. [55]	Porcine model of volume controlled mechanical ventilation (*n* = 5)	Total particle count	(1) Total particle count at a PEEP level of 15 cmH_2_O was lower than that of 5 cmH_2_O (282 vs. 3754, *p* < 0.009).
Carpaij et al. [56]	Asthma (*n* = 46) and control (*n* = 18)	Particle mass	(1) PExA mass was significantly lower in persistent asthma compared to complete asthma remission and control subjects (*p* = 0.028 and *p* = 0.003, respectively). (2) PExA mass was significantly lower in clinical asthma remission compared to control subjects (*p* = 0.018).
Elimsson et al. [57]	Chronic non-productive cough (*n* = 14)	Proteins	(1) Proteomic analysis showed 75 proteins significantly altered in patients with chronic cough compared to those in control (*p* < 0.05) involved in immune and inflammatory responses, complement and coagulation system, epithelial junction integrity proteins, and in neuroinflammatory responses.
Emilsson et al. [58]	Gastroesophageal reflux, asthma, and bronchitis (*n* = 48)	SP-A and albumin	(1) SP-A (25 vs. 38 mg/g PEx, *p* < 0.001) and albumin (48 vs. 73 mg/g PEx, *p* < 0.001) in PEx were lower among gastroesophageal reflux subjects than those in controls.
Ericson et al. [59]	Lung transplant recipients: control (*n* = 26) vs. bronchiolitis obliterans syndrome (*n* = 7).	Total particle count, SP-A, albumin	(1) Lung transplant recipients exhaled higher numbers of particles (8 vs. 1.8 ng/L, *p* < 0.0001) than controls. (2) SP-A in exhaled particles and the SP-A/albumin ratio were lower (18 vs. 30 mg/mL, *p* = 0.002; 0.35 vs. 0.74, *p* = 0.0001) in the bronchiolitis obliterans syndrome (BOS) group compared to those in the BOS-free group.
Hallgren et al. [60]	Elective open-heart surgery receiving mechanical ventilation (*n* = 30)	Particle flow rate	(1) Ventilation with pressure-regulated volume control (PRVC) resulted in the lowest PFR compared to VCV (*p* = 0.0285) and PCV (*p* = 0.0149). (2) Ventilation with pressure support ventilation (PSV) resulted in significantly higher PFR (2249 ± 426 particles/min) compared to all other ventilation modes used.
Hirdman et al. [61]	COVID-19 (*n* = 29)	Exhaled breath particles	(1) There was a significant increase in particles per exhaled volume in COVID-positive patients compared to those in healthy controls (*p* < 0.001). (2) Pulmonary surfactant-associated protein B (SFTPB, E) was significantly downregulated in COV-POS and COV-NEG (symptomatic) patients versus that in the healthy control group.
Holz et al. [25]	Segmental and inhalation endotoxin challenge in healthy volunteers (*n* = 10)	Concentrations of IL-6 and IL-8 per ng PExA	(1) Clear increase in the concentrations of IL-6 5 h post-segmental (*p* < 0.001) and post-inhalation LPS challenge (*p* < 0.001) was detected. (2) Clear increase in the concentrations of IL-8 5 h post-segmental (*p* < 0.01) and post-inhalation LPS challenge (*p* < 0.001) was detected.
Hussain-Alkhateeb et al. [19]	Asthma (*n* = 16) and smokers (*n* = 17)	Phospholipids, SP-A, albumin	(1) The phospholipids (PC14:0/16:0 and PC16:0/18:2) and SP-A were higher, and albumin was lower among the subjects with asthma. (2) Higher levels of DPPC observed in smokers compared to non-smokers.
Koca et al. [39]	Healthy volunteers (*n* = 97) and smokers (*n* = 15)	SP-A	(1) No correlation between PEx and plasma SP-A levels (*p* = 0.15) in healthy participants. (2) The ratio of plasma to PEx SP-A significantly higher in current smokers compared to that in healthy participants (*p* = 0.003).
Kokelj et al. [47]	Current smokers (*n* = 38), former smokers (*n* = 47), healthy controls (*n* = 22)	Proteins in PExA samples	(1) Eighty-one proteins altered in current smokers compared to those in never-smokers (*p* < 0.05). (2) Relative abundance of 58 proteins significantly altered in female current smokers as compared to those in non-smokers (*p* < 0.05), while 27 proteins significantly altered in male current smokers (*p* < 0.05). (3) Protein alterations consistent with complement pathway activation in female smokers.
Kokelj et al. [46]	Asthma (*n* = 20)	Complement and coagulation proteins	(1) Nine proteins were differentially abundant in subjects with asthma as compared to controls. (2) C3 was significantly higher in inadequately controlled asthma as compared to that in well-controlled asthma.
Larsson et al. [13]	Asthma (*n* = 13)	Mass of exhaled particles, SP-A, albumin	(1) Total mass of exhaled particles was lower in the asthma patients (900 pg/L of exhaled air) compared to that in control (1710 pg/L of exhaled air) during pollen season. (2) No significant effect on the concentration of SP-A and albumin in exhaled particles.
Larsson et al. [48]	Healthy participants with segmental/inhalation LPS challenge (*n* = 10)	Phospholipid composition of PEx	(1) The overall phospholipid composition of BAL, ISP, and PEx was similar, with PC (32:0) and PC (34:1) representing the largest fractions in all three sample types. (2) An increase of SM (d34:1) following segmental LPS challenge was detectable in PEx.
Lärstad et al. [14]	COPD (*n* = 13)	Total particle count, SP-A	(1) COPD patients had lower particle number concentration than healthy subjects (*p* < 0.0001). (2) COPD patients exhibited significantly lower SP-A mass content of the exhaled particles (2.7 vs. 3.9 wt%, *p* = 0.036).
Lindstedt and Hyllen [62]	Cardiac failure with mechanical ventilation (*n* = 10) vs. control (*n* = 10)	Particle flow rate	(1) Median PFR in patients with cardiac failure higher than PFR in patients with normal cardiac function (*p* < 0.001). (2) Median particle mass greater in the cardiac failure group compared to that in the control group (*p* = 0.002). (3) Patients with post-operative cardiac failure following cardiac surgery exhibit an increase in exhaled particles mass and PFR compared with the control group.
Ljungkvist et al. [63]	Stainless steel welders (*n* = 19)	Metals	(1) All samples, including blanks, had quantifiable amounts of metals; however, no statistically significant increase in the analysed metals in PExA over the working shift (*p* = 0.6 for chromium, manganese, and nickel).
Östling et al. [37]	Asthma (*n* = 20) and healthy controls (*n* = 10)	Proteins in PExA samples	(1) A total of 207 proteins were detected in up to 80% of the PExA samples.
Soares et al. [44]	Asthma (*n* = 83) and healthy volunteers (*n* = 32)	SP-A, albumin	(1) PExA method has the potential to non-invasively sample small airways-derived proteins (SP-A and albumin) associated with airway dysfunction phenotypes in asthma. (2) Modest but significant correlations were found for %SP-A with oscillometry parameters of small airway dysfunction. (3) Albumin demonstrated a significant correlation with FVC and GINA treatment (*p* < 0.05).
Stenlo et al. [64]	Porcine model with LPS-induced ARDS (*n* = 7)	Particle flow rate	(1) The particle flow rate increased significantly over time after LPS administration, from baseline (*p* = 0.0012) to after 60 min in all 7 animals.
Viklund et al. [65]	Current smokers (*n* = 37) and never-smokers (*n* = 29)	Total particle count, phospholipids, SP-A	(1) Smoking increased the exhaled number of particles (20.8 vs. 13.2 kn/L, *p* = 0.011). (2) Smoking increased contents of DPPC (11.3 vs. 10.3 wt%, *p* = 0.025) and POPC (3.7 vs. 2.9 wt%, *p* = 0.008). (3) Smoking increased contents of SP-A (3.9 vs. 3.1 wt%, *p* = 0.037).
Zwitserloot et al. [66]	Cystic fibrosis (*n* = 23)	Particles in exhaled air mass and number	(1) Correlation between lung clearance index and PEx ng/l was low (*p* = 0.07). (2) PExA device is feasible to use in children; however, it is a less sensitive tool to detect small airway diseases as it does not differentiate healthy children from children with cystic fibrosis.

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
