# Peer review of "Particles in Exhaled Air (PExA): Clinical Uses and Future Implications"

_diagnostics, 2024, doi:10.3390/diagnostics14100972_

Round 1
Reviewer 1 Report
Comments and Suggestions for Authors
I read this review with great interest. The review demonstrates a deep understanding of respiratory diagnostics and the implications of new technologies like PEXA. It is evident that the reviewer has a strong background in the field, which lends credibility and depth to the feedback provided. The review is constructive and very well-written.
I found several major concerns which need to be addressed before publication:
- A clearer explanation of the sample sizes used in various studies cited is needed, as well as the selection criteria for participants. This information is crucial for assessing the validity and applicability of the study results.
- Authors should discuss the steps taken to standardize the PEXA procedure and ensure reproducibility of the results across different settings and populations. This is important for the technology adoption in routine clinical practice.
- The paper mentions various traditional and novel methods of respiratory samplin, but a more detailed comparative analysis with PEXA, highlighting specific advantages and limitations, would provide a clearer value proposition for the PEXA method.
- Info on the long-term reliability and maintenance requirements of the PEXA device would be useful for potential adopters.
Author Response
We are very grateful for the reviewer’s comments. Please see our detailed response to these comments.
Reviewer 1
- A clearer explanation of the sample sizes used in various studies cited is needed, as well as the selection criteria for participants. This information is crucial for assessing the validity and applicability of the study results.
- Thank you for your detailed observation. We have now added sample size to all the studies mentioned in the text. Sample size for the studies included in the table has already been presented.
- Authors should discuss the steps taken to standardize the PEXA procedure and ensure reproducibility of the results across different settings and populations. This is important for the technology adoption in routine clinical practice.
- We have now explicitly stated how procedures of standardisation occur for both breathing and analytical techniques through protocolisation. This is detailed in section 4, PExA techniques. Studies detailing the methodologies are referenced appropriately.
- The paper mentions various traditional and novel methods of respiratory sampling, but a more detailed comparative analysis with PEXA, highlighting specific advantages and limitations, would provide a clearer value proposition for the PEXA method.
- The limitations of each sampling technique have been discussed as part of their description and the question posted at the end of “RTFL Sampling” section has been more explicitly answered within the “PExA Basics” section.
- We have now included additional sentences (lines 214-212) to include further details of studies comparing PExA with BALF.
- Info on the long-term reliability and maintenance requirements of the PEXA device would be useful for potential adopters.
- We have introduced a new section to address this comment (lines 331-345), 4.3: Maintenance Requirements of PExA Machine)
Reviewer 2 Report
Comments and Suggestions for Authors
The article “Particles in Exhaled Air (PExA): Clinical Uses and Future Implications” is devoted to the development of non-invasive methods for diagnosing diseases involved in the pathogenesis of airway inflammation. Accurate diagnosis of respiratory diseases is critical given the increasing number of patients with chronic respiratory diseases. It is obvious that the proposed development has significant potential, which can be realized using systems biology approaches based on the analysis of biomarkers of asthma and COPD. Discuss and cite the articles:
Hachim, M.Y.; Alqutami, F.; Hachim, I.Y.; Heialy, S.A.; Busch, H.; Hamoudi, R.; Hamid, Q. The Role of Systems Biology in Deciphering Asthma Heterogeneity. Life 2022, 12, 1562. https://doi.org/10.3390/life12101562
sbv IMPROVER project team (in alphabetical order), Boue S, Fields B, et al. Enhancement of COPD biological networks using a web-based collaboration interface. F1000Res. 2015;4:32. Published 2015 Jan 29. doi:10.12688/f1000research.5984.2
Sircar G, Saha B, Bhattacharya SG, Saha S. Allergic asthma biomarkers using systems approaches. Front Genet. 2014;4:308. Published 2014 Jan 8. doi:10.3389/fgene.2013.00308
Author Response
We are very grateful for the reviewer’s comments. Please see our detailed response to these comments.
Reviewer 2
Discuss and cite the articles:
Hachim, M.Y.; Alqutami, F.; Hachim, I.Y.; Heialy, S.A.; Busch, H.; Hamoudi, R.; Hamid, Q. The Role of Systems Biology in Deciphering Asthma Heterogeneity. Life 2022, 12, 1562. https://doi.org/10.3390/life12101562
sbv IMPROVER project team (in alphabetical order), Boue S, Fields B, et al. Enhancement of COPD biological networks using a web-based collaboration interface. F1000Res. 2015;4:32. Published 2015 Jan 29. doi:10.12688/f1000research.5984.2
Sircar G, Saha B, Bhattacharya SG, Saha S. Allergic asthma biomarkers using systems approaches. Front Genet. 2014;4:308. Published 2014 Jan 8. doi:10.3389/fgene.2013.00308
- The advent of multi-omics into standard clinical practice is truly the next frontier of medicine. We have included all these references within the “Future Directions” section (lines 580-584).